# Grade Discovery as an Identifiability Diagnostic for Clifford-Valued Features

## Abstract

Clifford-valued models can store scalars, vectors, bivectors and pseudoscalars in a common algebra, but the geometric type of each feature channel is usually specified before training. We do not argue that all such choices require discovery. Many choices are obvious from physical meaning, such as mass as a scalar or velocity as a vector, and other choices are less visible from coordinate dimension alone because a polar vector and the axial vector coordinates of a bivector have the same number of coordinate components while transforming differently under reflections. We study *grade discovery* as an identifiability diagnostic for such cases. Given paired observations before and after known or estimated geometric transformations, the diagnostic fits a soft weight over a finite set of dimension-matched candidate transformation laws, meaning candidates with the same number of coordinate components, by minimising an equivariant least-squares loss. The classical representation-theoretic fact behind the problem is that rotations cannot distinguish scalar from pseudoscalar or vector from bivector in three-dimensional space, whereas reflections can. Our contribution is an auditable machine learning test built from this fact. We prove that the expected noiseless loss obtained by replacing finite averages with expectations over features and transformations, identifies the true candidate exactly when the observed transformations separate the candidate laws and the conditional second moment matrix is positive definite on the tested coordinate block. Controlled experiments validate this separation principle. Real geometry experiments on MD17 ethanol recover force as a vector and a torque-like cross-product channel as a bivector. Landmark registration experiments estimate the transformation matrix by Kabsch alignment from perturbed paired atom positions and quantify operational breakdown thresholds at which transformation estimation breaks the diagnostic. A three-layer grade structured $O(3)$-equivariant message-passing experiment, inspired by Clifford architectures shows that the discovered grade assignment can change downstream model design and reduce reflection test error relative to incorrect hard-coded grade choices.

## 1 Introduction

Geometric machine learning models encode how features transform when the input is rotated, reflected, translated or otherwise acted on by a symmetry group (Bronstein et al., 2021). This design is useful because it makes a learned map respect known symmetries by construction, and it has led to typed feature fields in group equivariant, steerable, Euclidean, and gauge equivariant networks (Cohen & Welling, 2016; Weiler & Cesa, 2019; Fuchs et al., 2020; Satorras et al., 2021; Geiger & Smidt, 2022; Cohen et al., 2019). Clifford and geometric algebra models use this idea in a compact form, since a single multivector can contain scalar, vector, bivector, and higher-grade components (Doran & Lasenby, 2003). Building on this representation, Clifford neural layers, Geometric Clifford Algebra Networks (GCANs), Clifford Group Equivariant Neural Networks (CGENNs), metric learning for CGENNs and the Geometric Algebra Transformer (GATr) provide expressive architectures for geometric data (Brandstetter et al., 2023; Ruhe et al., 2023b;a; Ali et al., 2024; Brehmer et al., 2023).

Although these architectures make grade structured computation possible, they usually require the practitioner to specify feature grades before training. We agree that this requirement is often benign and in many

Table 1: Examples of grade choices where the diagnostic is usually unnecessary and where it is informative. Chirality means handedness and signed volume quantities are orientation sensitive scalars, for example scalar triple products, whose sign changes under reflections.

| Feature situation | Usual geometric type | Role of the diagnostic |
|---|---|---|
| Atom type, mass, temperature | Scalar | Usually unnecessary because the grade follows from the data description. |
| Position, displacement, velocity, force | Polar vector | Usually unnecessary when the physical meaning is known. |
| Torque, angular momentum and cross products such as $r \times F$ ($r$ is a centred position, $F$ is a force) | Axial vector or bivector coordinates | Useful when a channel is supplied as three numbers but parity metadata are unclear. |
| Chirality or signed volume quantities | Pseudoscalar | Useful only if reflections are observed or estimated reliably. |

cases no diagnostic is needed. Some assignments follow directly from the data description, since an atom type or temperature is naturally scalar and a force or velocity is naturally a polar vector. However, there are also common situations in which two candidate types have the same coordinate dimension but differ only in parity. In three dimensions, a force and a torque-like quantity can both be stored using three real coordinate components, yet the former is a polar vector while the latter is an axial vector, equivalently the Hodge-dual coordinate representation of a bivector. Consequently, a rotation-only protocol gives no evidence that can separate these two cases, whereas a protocol that includes reflections can expose the parity difference.

Table 1 makes this distinction explicit. The diagnostic is not meant to replace obvious physical typing. It is meant for same dimensional alternatives where the coordinate count alone does not determine the transformation law. We therefore do not claim that ordinary physical quantities generally need to be discovered. The intended use case is an ambiguous, derived, learned or metadata-poor channel whose coordinate dimension is compatible with more than one Clifford grade.

This paper asks when such typing is supported by transformation evidence rather than only by a modelling convention. The question is narrower than general symmetry discovery because we do not try to learn an unknown group from raw observations. Instead, we assume that paired transformed observations are available, either from controlled augmentation, physical or simulation probing or transformation estimation from landmarks, and we ask whether those observations identify a finite candidate grade assignment (finite means a predefined list such as {vector, bivector} for a three-coordinate block, rather than an unrestricted space of possible representation laws).

> **Research question.** Given paired observations of a feature channel before and after known or estimated geometric transformations, when is its Clifford grade identifiable from the data and when is the grade assignment only a modelling assumption?

We address this question by formulating *grade discovery* as a finite candidate, dimension-matched diagnostic. Dimension-matched means that the candidate laws have the same number of coordinate components, for example vector and bivector laws acting on a three-coordinate block. For each feature channel, the method compares candidate representation matrices of the same coordinate dimension and fits a soft mixture of those candidates using an equivariant prediction loss. This formulation intentionally isolates identifiability from the optimisation and capacity issues of a large downstream model. The paper makes four contributions. First, it gives a least-squares formulation with a differentiable softmax gate that can be embedded in neural training. Second, it proves that the diagnostic identifies a grade exactly when the observed transformations separate the candidate representation laws on non-degenerate data (the conditional second moment matrix is positive definite on the coordinate block being compared). Third, it expands the empirical evidence beyond ideal synthetic pairs by using MD17 molecular geometries (Fey & Lenssen, 2019) and a noisy landmark Kabsch estimator for the transformation matrix (Kabsch, 1976) (we perturb paired landmarks, align them by orthogonal Procrustes/Kabsch and then use the estimated transformation matrix in the diagnostic). Fourth, it tests practical model selection with a three-layer grade structured $O(3)$-equivariant message-

passing network inspired by Clifford architectures, where the discovered assignment performs close to the true assignment and incorrect parity assignments fail on reflection containing tests.

The scope is important. We do not claim to discover arbitrary unknown group representations, infer the ambient coordinate dimension or decompose an unconstrained mixed multivector without a candidate block structure. The diagnostic answers a conditional question. Given a coordinate block and a finite set of candidate Clifford transformation laws, it reports whether the transformations present in the data support one candidate, leave the block ambiguous or make the supplied transformation metadata (the recorded or estimated transformation matrix for a paired sample including its determinant sign) unreliable.

## 2 Related Work

**Equivariant models with prescribed representation types.** Here 'representation type' means the rule by which a feature field transforms, for example as a scalar, vector, tensor or irreducible representation channel (a feature subspace that transforms according to one irreducible group representation). Equivariant networks typically assign each feature field to a representation of the symmetry group, after which the architecture restricts its layers to respect those representations (Cohen & Welling, 2016; Weiler & Cesa, 2019; Cohen et al., 2019). In three-dimensional learning, Tensor Field Networks, SE(3)-Transformers, E(n)-equivariant Graph Neural Networks (EGNNs) and e3nn represent different points in the design space between explicit higher-order representations, attention mechanisms, coordinate updates and software systems for Euclidean equivariance (Thomas et al., 2018; Fuchs et al., 2020; Satorras et al., 2021; Geiger & Smidt, 2022). These methods motivate the central premise that representation type matters. Our diagnostic is not a new equivariant layer and it audits whether a proposed type assignment is supported by paired transformation evidence.

**Clifford-valued architectures.** Clifford algebra organises geometric quantities into grades within a common algebra (Doran & Lasenby, 2003). Recent machine learning architectures use this structure for partial differential equation (PDE) surrogates with Clifford convolutions (Brandstetter et al., 2023), group action layers in GCANs (Ruhe et al., 2023b), Clifford group equivariant layers that preserve multivector grading (Ruhe et al., 2023a) and projective geometric algebra transformers for heterogeneous geometric objects (Brehmer et al., 2023). Metric learning for CGENNs makes another algebraic modelling choice data-driven, namely the metric used inside the Clifford construction (Ali et al., 2024). Grade discovery is complementary to these approaches because it does not replace grade-aware architectures, but checks whether a channel's assigned grade is identifiable from the transformations that the data expose.

**Symmetry and representation discovery.** Several methods aim to infer symmetries or latent group structure from data. LieGAN learns continuous symmetry transformations through an adversarial objective (Yang et al., 2023), SymmetryGAN studies automatic discovery of distributional symmetries (Desai et al., 2022) and group structured representation learning seeks disentangled latent factors in dynamical environments (Quessard et al., 2020). These approaches address broader discovery problems than ours, since they aim to infer transformations or latent group structure. They also answer a different algebraic question. Discovering a continuous rotation symmetry can reveal that a dataset is compatible with proper rotations, but it does not by itself decide whether a three-coordinate channel is a polar vector or an axial bivector, because these two candidate laws coincide on proper rotations. The ambiguity is discrete and parity-sensitive rather than a failure to find a continuous generator. Thus even a successful symmetry discovery stage that recovers an $SO(3)$ action is structurally blind to the parity distinction that separates $SO(3)$ from $O(3)$. *This is not a weakness of those methods, but a different identifiability question.* In contrast, grade discovery assumes a known or estimated transformation matrix and asks which member of a finite Clifford candidate set matches how a channel transforms. This narrower diagnostic role is deliberate. Our method cannot replace general symmetry discovery, but it can diagnose grade assignments inside an existing equivariant modelling pipeline.

Table 2 makes the comparison. Grade discovery is a finite candidate detection rather than general representation discovery. Its role is not to infer an unknown symmetry group or an unrestricted latent representation. Its role is to make a Clifford modelling decision auditable after a practitioner has specified a coordinate block, a transformation source and a finite set of candidate grade laws. The output is not merely a class label. It

Table 2: Positioning relative to symmetry discovery and representation learning methods.

| Method class | Typical output | Relation to grade discovery |
|---|---|---|
| Symmetry discovery such as LieGAN and Symmetry-GAN (Yang et al., 2023; Desai et al., 2022) | Transformations or symmetry structure of a dataset | Broader problem. We do not infer the group or the transformation family. |
| Latent group structure learning such as Quessard et al. (Quessard et al., 2020) | Disentangled latent factors or group structure in dynamics | Broader problem. We do not learn arbitrary latent representations. |
| Typed equivariant and Clifford-valued networks | Architectures using prescribed representation types | Our diagnostic audits one part of this prescription. |
| Grade discovery in this paper | Identifiability, ambiguity, or failure of a declared coordinate block within a finite Clifford candidate set | Narrower problem. The benefit is an auditable yes, no or cautionary answer under supplied transformation evidence. |

is an identifiability statement within the supplied candidate set, i.e. the diagnostic can report support for one candidate, ambiguity because the transformations do not separate the candidates or caution because the transformation estimate appears unreliable. This narrowness is a conceptual strength for the present diagnostic, because the method can return a mathematically interpretable negative answer. For example, if the available evidence contains only proper rotations, then no optimisation method can justify choosing between a vector and a bivector law in the axial-coordinate representation. The diagnostic therefore separates two questions that are often conflated, i.e. whether a transformation family has been observed and whether that family contains enough parity evidence to identify a Clifford grade. In this sense, *the question addressed here is downstream of general symmetry discovery, i.e. once a transformation family or an estimated transformation matrix is available, does that evidence separate the Clifford candidate laws used by the model?*

## 3 Problem Setup and Diagnostic

Let $V$ denote three-dimensional Euclidean space and let $\mathbb{R}^3$ denote its coordinates in an orthonormal basis. The real Clifford algebra over $V$ is $\mathrm{Cl}_3$. A grade-zero element is a scalar, a grade-one element is a vector, a grade-two element is a bivector and a grade-three element is a pseudoscalar. We represent bivectors with the Hodge-dual axial basis $B_1 = e_2 e_3$, $B_2 = e_3 e_1$ and $B_3 = e_1 e_2$, so bivectors have three coordinates like vectors. Let $Q \in O(3)$ be an orthogonal matrix, where $O(3)$ contains rotations and reflections, and let $\det(Q)$ denote its determinant. The subgroup $SO(3)$ contains the rotations which are the orthogonal matrices with determinant one. We write $\rho_t(Q)$ for the representation matrix associated with candidate type $t$. The candidate representation matrices are:

$$\begin{aligned} \rho_{\mathrm{scal}}(Q) &= 1, \quad \rho_{\mathrm{pscal}}(Q) = \det(Q), \\ \rho_{\mathrm{vec}}(Q) &= Q, \quad \rho_{\mathrm{biv}}(Q) = \det(Q)Q \end{aligned} \tag{1}$$

The first two laws act on one-dimensional coordinates, and the last two laws act on three-dimensional coordinates. Therefore the dimension-matched candidate sets are:

$$\mathcal{C}_1 = \{\mathrm{scal}, \mathrm{pscal}\}, \qquad \mathcal{C}_3 = \{\mathrm{vec}, \mathrm{biv}\} \tag{2}$$

The diagnostic intentionally excludes cross-dimensional comparisons such as a one-dimensional scalar law versus a three-dimensional vector law. In the present formulation, those laws act on different coordinate spaces and cannot be mixed in a single prediction law without introducing an additional embedding or projection. If the coordinate block is already observed, a one-coordinate block and a three-coordinate block are separated by shape. This is a deliberate scope restriction rather than an attempted solution to embedding dimension discovery. The nontrivial ambiguity studied here is different, i.e. the candidate matrices have

the same shape but agree on some transformation families and differ only through parity. Equation 1 is the classical parity distinction behind the diagnostic. If $Q \in SO(3)$, then $\det(Q) = 1$, so scalar and pseudoscalar laws coincide and vector and bivector laws coincide in the axial-coordinate representation. If $Q$ is a reflection, then $\det(Q) = -1$, so both pairs are separated by a sign.

The diagnostic is a single-grade or blockwise test. Each diagnostic channel is assumed to be a homogeneous coordinate block with a declared coordinate dimension. For multivector inputs used in GCANs, CGENNs or GATr, the same test can be applied to declared one-dimensional or three-dimensional coordinate blocks or to block diagonal candidate laws when the block partition is already known (for example, when a multivector is stored as separately declared scalar, vector, bivector and pseudoscalar coordinate blocks). It does not infer the partition itself, decide whether a raw channel should be one- or three-dimensional, or unmix an arbitrary dense multivector without additional assumptions.

**Assumption 1** (Blockwise homogeneous candidates). *Each diagnostic unit is a coordinate block whose candidate laws act on the same coordinate space. For multivector inputs, the block partition is assumed to be known and the diagnostic either compares candidate laws for one block at a time or compares a finite set of block diagonal assignments built from those declared blocks.*

For a channel $c$, let $d_c \in \{1,3\}$ be its coordinate dimension and let $x_{cj} \in \mathbb{R}^{d_c}$ be the untransformed feature for item $j$. For transformation index $s$, the observed transformed feature is $y_{csj} \in \mathbb{R}^{d_c}$ and the known or estimated orthogonal matrix is $Q_s$. A candidate type $t \in \mathcal{C}_{d_c}$ has representation matrix $\rho_t(Q_s)$. The diagnostic assigns non-negative weights $p_{ct}$ that sum to one over $t \in \mathcal{C}_{d_c}$, and it forms the mixed representation $R_c^{p_c}(Q_s) = \sum_{t \in \mathcal{C}_{d_c}} p_{ct} \rho_t(Q_s)$.

Let $S$ be the number of transformations and let $N_c$ be the number of examples for channel $c$. The channel loss is the mean squared prediction error:

$$\ell_c(p_c) = \frac{1}{SN_c} \sum_{s=1}^{S} \sum_{j=1}^{N_c} \|y_{csj} - R_c^{p_c}(Q_s) x_{cj}\|_2^2 \tag{3}$$

We report this prediction error in the experiments as mean squared error (MSE). When the diagnostic is optimised by gradient descent, we train unconstrained logits $\alpha_{ct}$ and define the probabilities by:

$$p_{ct} = \frac{\exp(\alpha_{ct})}{\sum_{u \in \mathcal{C}_{d_c}} \exp(\alpha_{cu})} \tag{4}$$

Thus $p_{ct}$ is not an independent unconstrained parameter. It is the softmax normalised value of $\alpha_{ct}$. In the two-candidate experiments (scalar versus pseudoscalar or vector versus bivector) we also use the closed-form least-squares solution derived in Appendix B, because this removes optimiser noise (variation due to finite step gradient descent, learning rate choices or random logit initialisation) from the identifiability measurements. Paired evidence (an untransformed feature block, the same block after a geometric action, and the corresponding matrix $Q_s$ or estimate $\widehat{Q}_s$) can arise in three regimes. In augmentation pairs, $Q_s$ is chosen by the practitioner and is exactly known. In physical or simulation probing, the same system is transformed under a controlled transformation. In registration pairs, $Q_s$ is estimated from paired landmarks or point clouds. The diagnostic is strongest in the first two regimes and becomes conditional on transformation estimation quality in the third. It should not be applied to raw observational data without either transformation metadata or a reliable registration stage. Table 3 summarises how the experiments instantiate these evidence regimes and which claims each one supports.

## 4 Identifiability Analysis

The diagnostic is useful only if its optimum (the minimiser of the loss over the probability simplex of candidate weights) has a clear interpretation. We analyse a noiseless population setting (the infinite-sample idealisation in which the empirical average in Eq. 3 is replaced by an expectation over $x$ and $Q$, and observation noise is absent). Let $Q$ be a random observed transformation, let $x \in \mathbb{R}^d$ be a random feature vector and let $t_\star \in \mathcal{C}_d$ be the true candidate. Define the conditional second moment matrix $G(Q) = \mathbb{E}[xx^\top \mid Q]$. We assume that

Table 3: Scope and evidence in the experiments. The table summarises which question each experiment answers and which claim it can support.

| Experiment | Data or model | Transformation evidence | Supported claim |
|---|---|---|---|
| Controlled diagnostic | Synthetic scalar, pseudoscalar, vector and bivector channels | Exact rotations and reflections | Validates the separation theorem and finite-sample reflection effect. |
| Real geometry | MD17 ethanol positions and forces | Exact augmentation and estimated $Q$ | Tests the diagnostic on non-Gaussian molecular feature distributions. |
| Estimated $Q$ | MD17 landmarks with added landmark noise | Kabsch-estimated orthogonal matrices | Quantifies degradation when transformation metadata must be inferred. |
| Downstream model | Three-layer grade structured $O(3)$-equivariant message-passing | Same architecture with different grade assignments | Tests whether discovered grades change a deeper model design. |
| Hidden probe | Saved downstream hidden fields | Paired hidden activations under $O(3)$ | Checks declared intermediate fields, without claiming arbitrary latent representation discovery. |

$G(Q)$ is positive definite almost surely on the candidate coordinate block. In the augmentation experiments below, $Q$ is sampled independently of $x$, so this condition reduces to $G = \mathbb{E}[xx^\top] \succ 0$. The transformed feature is $y = \rho_{t_\star}(Q)x$.

**Theorem 1** (Finite candidate identifiability). *Let $\mathcal{C}_d$ be a finite set of candidate representation laws with common coordinate dimension $d$. If the only probability vector $p$ satisfying:*

$$\sum_{t \in \mathcal{C}_d} p_t \rho_t(Q) = \rho_{t_\star}(Q) \quad \text{almost surely} \tag{5}$$

*is the one-hot vector on $t_\star$, then the population version of Eq. 3 (the infinite-sample expected loss obtained by replacing the empirical average over examples and transformations with an expectation) has a unique zero-loss minimiser at the true-type. Conversely, if two candidate laws are equal almost surely on the observed transformations, then the population loss cannot distinguish those two candidates. Here 'almost surely' has its standard probabilistic meaning, so the equality may fail only on a set of transformations with probability zero under the observed transformation distribution. The condition excludes singular or degenerate transformation protocols that do not provide sufficient parity-sensitive evidence.*

The proof uses the residual matrix $M_p(Q) = \rho_{t_\star}(Q) - \sum_{t \in \mathcal{C}_d} p_t \rho_t(Q)$, so the population loss can be written as:

$$\mathcal{L}(p) = \mathbb{E}_Q \left[ \operatorname{tr} \left( M_p(Q)^\top M_p(Q) G(Q) \right) \right] \tag{6}$$

Equation 6 makes the two requirements explicit. The transformation distribution must make $M_p(Q)$ nonzero for incorrect mixtures, and the conditional second moment matrix $G(Q)$ must expose the directions where that matrix difference acts. The theorem is therefore not an optimiser convergence statement, it is a statement about the null space of the representation differences under the observed transformation distribution and the conditional second moment matrix. The theorem formalises a simple principle. The loss can identify a grade only through differences in the candidate predictions that are actually exposed by the observed transformations and the conditional second moment matrix. Because this principle is representation-theoretic rather than optimiser specific, it applies equally to the closed-form estimator and to the differentiable softmax gate. For the two-candidate cases used in the experiments, the amount of separating evidence can be read directly from the least-squares denominator. If candidates $a$ and $b$ are compared on samples $\{(x_j, Q_s)\}$, define:

$$E_{ab} = \sum_{s,j} \|(\rho_a(Q_s) - \rho_b(Q_s)) \, x_j\|_2^2 \tag{7}$$

A zero value means that the two candidate laws make identical predictions on the observed evidence, while a positive value means that the data expose at least some separating direction. Appendix B shows that this denominator is exactly the normalising term in the closed-form two-candidate estimator, and Appendix I uses it to interpret the controlled stress tests.

**Proposition 1** (Parity-sensitive pairs in three dimensions)**.** *For the representations in Eq. 1, rotations alone do not identify scalar versus pseudoscalar or vector versus bivector. If the observed transformation distribution satisfies $\Pr[\det(Q) = -1] > 0$ and $G(Q)$ is positive definite almost surely on the candidate block, then both pairs are identifiable in the population setting.*

**Corollary 1** (Finite-sample probability of observing parity evidence)**.** *Suppose each of the $S$ sampled transformations is a reflection with probability $r$, independently across transformations. The probability that the sample contains at least one parity revealing transformation is:*

$$P_{\text{sep}}(r, S) = 1 - (1 - r)^S \tag{8}$$

*If a no-reflection sample is reported as the uninformative weight $1/2$ and a separated noiseless sample is reported as weight one on the true-type, the expected reported true-type weight is:*

$$W_{\text{true}}(r, S) = 1 - \frac{1}{2}(1 - r)^S \tag{9}$$

The corollary should not be read as saying that a nonzero reflection fraction produces a fractional population optimum. In the population setting, any positive reflection probability separates the parity-sensitive pairs. The probability $P_{\text{sep}}(r, S)$ in Eq. 8 describes the chance that a diagnostic run has observed at least one separating transformation, while $W_{\text{true}}(r, S)$ in Eq. 9 is the corresponding expected reported true-type weight under the $1/2$-versus-$1$ reporting convention. Formal proofs and algebraic derivations are given in Appendices A to E. In particular, Appendix C proves Theorem 1 and Appendix D proves Proposition 1.

**Remark 1** (Finite candidate sets and blockwise mixed inputs)**.** *The theorem is stated for an arbitrary finite candidate set of common coordinate dimension. The experiments use pairs because scalar/pseudoscalar and vector/bivector are the same dimensional parity ambiguities in $\text{Cl}_3$. If a multivector input has a known block partition, one can apply the diagnostic to each block or compare block diagonal candidate laws. This is an extension of the finite candidate formulation, not an algorithm for discovering the block partition itself.*

**Corollary 2** (Blockwise finite candidate products)**.** *Suppose a multivector feature is partitioned into declared coordinate blocks $b = 1, \dots, B$. Let each block have a finite candidate set of common coordinate dimension and let a product assignment act by the block diagonal law obtained by placing the chosen block representation matrices on the diagonal. If the resulting finite set of block diagonal laws satisfies the separation condition in Theorem 1 under a positive definite conditional second moment matrix for the concatenated blocks, then the product assignment is identifiable. In particular, when blocks are tested independently and each block satisfies the theorem's separation and conditional second moment conditions, the full declared multivector assignment can be audited blockwise. This corollary expands the finite candidate formulation to mixed-grade multivectors with a known partition, but it still does not infer the partition or discover unknown representation laws.*

## 5 Experiments

**Controlled identifiability.** The first experiment keeps the controlled benchmark because it tests the theorem directly. Each channel is generated from one of the four laws in Eq. 1 and the diagnostic must choose between the dimension-matched candidates in Eq. 2. Rotations are sampled by normalising a four-dimensional Gaussian quaternion and converting it to a rotation matrix. A reflection sample is generated by composing the rotation with $\text{diag}(-1, 1, 1)$ and each transformation is reflected independently with probability $r$. Transformed observations are corrupted by independent Gaussian observation noise with standard deviation 0.02, so the reported mixture MSEs have a visible but small noise floor. Figure 1 shows that rotations alone keep the true-type weight at $1/2$, while rotations with reflections converge to the true-type once at least one reflection appears. The right panel confirms the expected reported $W_{\text{true}}$ in Eq. 9 when $S = 16$. These plots are sanity checks that the diagnostic reports ambiguity exactly when the representation laws are indistinguishable. Appendix I and Figure 7 show the corresponding scalar/pseudoscalar loss geometry, where rotations create a flat simplex of equivalent mixtures and reflections create a unique minimiser.

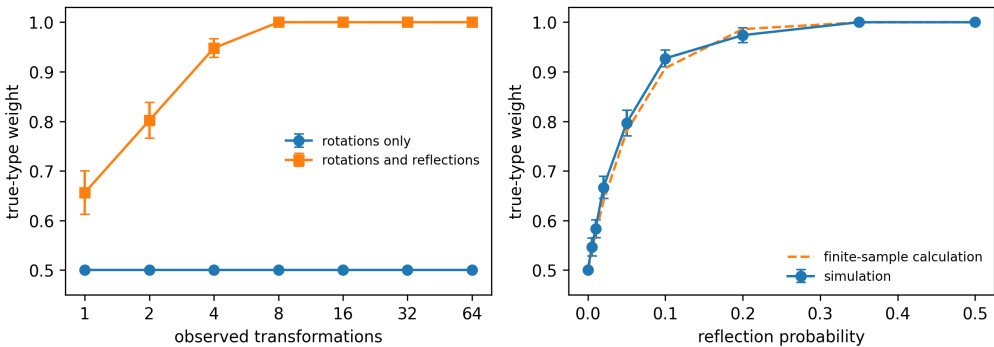

Figure 1: Controlled identifiability benchmark. Rotation-only evidence leaves parity-sensitive pairs at the uninformative true-type weight of $1/2$, whereas adding reflections creates separating evidence. The reflection-frequency curve shows the expected reported true-type weight $W_{\text{true}}$ which is induced by the finite-sample probability of seeing at least one reflection.

Table 4: Grade discovery on MD17 ethanol geometry. Values are means and standard errors over eight seeds. Exact rotations alone are insufficient, exact $O(3)$ transformations recover both channels, and estimated $Q$ remains effective when the determinant sign is recovered.

| Evidence | Channel | True law | $p_{\text{vec}}$ | True-type weight | Det. sign acc. |
|---|---|---|---|---|---|
| $SO(3)$ exact $Q$ | force | vector | $0.500 \pm 0.000$ | $0.500 \pm 0.000$ | $1.000 \pm 0.000$ |
| $SO(3)$ exact $Q$ | $r \times F$ | bivector | $0.500 \pm 0.000$ | $0.500 \pm 0.000$ | $1.000 \pm 0.000$ |
| $O(3)$ exact $Q$ | force | vector | $1.000 \pm 0.000$ | $1.000 \pm 0.000$ | $1.000 \pm 0.000$ |
| $O(3)$ exact $Q$ | $r \times F$ | bivector | $0.000 \pm 0.000$ | $1.000 \pm 0.000$ | $1.000 \pm 0.000$ |
| $O(3)$ estimated $Q$ | force | vector | $1.000 \pm 0.000$ | $1.000 \pm 0.000$ | $1.000 \pm 0.000$ |
| $O(3)$ estimated $Q$ | $r \times F$ | bivector | $0.000 \pm 0.000$ | $1.000 \pm 0.000$ | $1.000 \pm 0.000$ |
| estimated $Q$ forced to $SO(3)$ | force | vector | $0.500 \pm 0.000$ | $0.500 \pm 0.000$ | $0.520 \pm 0.021$ |
| estimated $Q$ forced to $SO(3)$ | $r \times F$ | bivector | $0.500 \pm 0.000$ | $0.500 \pm 0.000$ | $0.500 \pm 0.026$ |

**Real geometry diagnostic.** To test the diagnostic outside Gaussian synthetic coordinates, we use the MD17 ethanol CCSD(T) data loaded through PyTorch Geometric (Fey & Lenssen, 2019). The experiment uses 700 selected frames, split into 500 training frames and 200 test frames. Raw molecular coordinates are centred per frame. The augmentation matrices for MD17 are sampled from the same distribution as in the controlled experiment with quaternion sampled rotations optionally composed with $\text{diag}(-1, 1, 1)$, reflection probability 0 for $SO(3)$-only runs, and reflection probability 0.5 for $O(3)$ runs. The force $F_i \in \mathbb{R}^3$ at atom $i$ is treated as a polar vector and the torque-like quantity $\tau_i = r_i \times F_i$, where $r_i \in \mathbb{R}^3$ is the centred atom position, is treated as an axial vector, equivalently a bivector in Hodge-dual coordinates. Equivalently, $\tau_i$ is the Hodge-dual coordinate form of an antisymmetric $3 \times 3$ matrix, which behaves like a vector under proper rotations but acquires an additional determinant sign under reflections. This pair gives a physically grounded same-dimensional diagnostic case because both observed channels are $\mathbb{R}^3$-valued, yet their reflection laws differ. Table 4 reports the resulting grade weights. With exact rotations only, the diagnostic remains ambiguous. With exact rotations and reflections, it assigns weight one to force as a vector and to $r \times F$ as bivector. When $Q$ is estimated from noisy paired landmarks using the Kabsch orthogonal Procrustes estimator (Kabsch, 1976), the diagnostic still recovers both channels at the reference landmark noise level 0.02. This level corresponds to Gaussian landmark perturbations with standard deviation 0.02 times the empirical standard deviation of the transformed landmark coordinates, as detailed in Appendix F. When the estimator is forced to return an $SO(3)$ matrix even for reflected pairs, the determinant sign accuracy is approximately one half, because every reflected pair is forced to determinant $+1$, whereas proper rotation pairs retain the correct sign. This negative control shows that the method relies on reflection information in $Q$, not merely on a better optimiser or a favourable feature distribution.

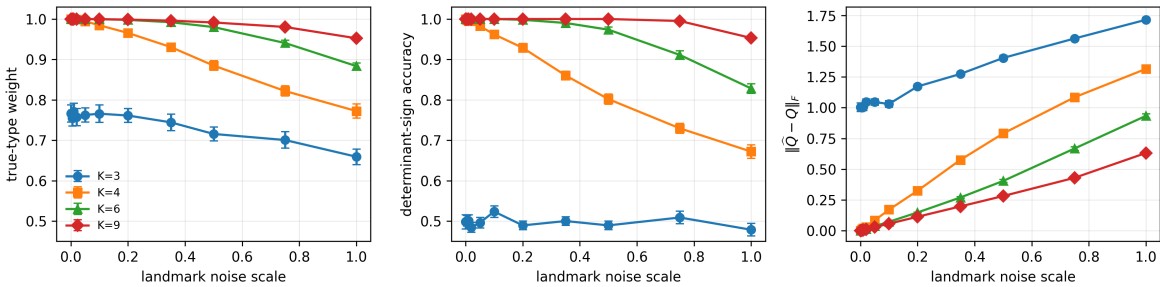

Figure 2: Estimated transformation degradation. Allowing $\widehat{Q} \in O(3)$ lets the diagnostic recover vector and bivector channels from noisy landmark pairs when determinant signs are estimated correctly, and failure appears first with too few landmarks or large landmark noise.

Table 5: Estimated-$Q$ threshold summary. The threshold is the first tested landmark noise scale at which the average true-type weight falls below 0.90. 'Not below' means the threshold was not reached in the tested range.

| Landmarks | Breakdown summary | Weight at max noise | Det. sign acc. at max noise |
|---|---|---|---|
| 3 | below 0.90 at 0 | $0.659 \pm 0.063$ | $0.479 \pm 0.014$ |
| 4 | below 0.90 at 0.5 | $0.772 \pm 0.050$ | $0.672 \pm 0.025$ |
| 6 | below 0.90 at 1 | $0.883 \pm 0.012$ | $0.828 \pm 0.004$ |
| 9 | not below 0.90 up to 1 | $0.952 \pm 0.002$ | $0.953 \pm 0.004$ |

**Estimated transformations from landmarks.** We next test what happens when the transformation matrix is estimated rather than given. In this molecular setting, the landmarks are the three-dimensional atom coordinates used for rigid registration. Figure 2 and Table 5 sweep landmark noise and the number of landmarks, denoted by $K$, used to estimate $\widehat{Q}$. A landmark noise scale $\sigma_{\text{lm}}$ means that each transformed landmark is perturbed by Gaussian noise with standard deviation $\sigma_{\text{lm}}$ times the empirical standard deviation of the transformed landmark coordinates. For $K < 9$, the $K$ atoms are sampled uniformly without replacement for each diagnostic pair. The result is conditional on the geometry of the registration problem. With three landmarks, the determinant sign is not reliably determined even without added noise, because three points in three dimensions are coplanar and do not define full handedness. With four landmarks, the average true-type weight falls below 0.90 by landmark noise scale 0.5. With six landmarks, it falls below 0.90 only at noise scale 1.0. With all nine ethanol atoms as landmarks, it remains above 0.90 up to the largest tested noise. Thus, estimated transformations are viable in this setting, but only when the paired landmarks are sufficiently informative to recover the reflection component. The three panels should be read together. The left panel is the grade discovery outcome, the middle panel shows whether the determinant sign was recovered and the right panel shows the Frobenius error of the estimated matrix. The main failure mode for grade discovery is not merely a large matrix error, but losing the determinant sign that separates vector from bivector under reflections. Table 5 therefore reports an operational breakdown threshold rather than a universal guarantee, i.e. it is the first tested noise level at which the average true-type weight falls below the stated threshold for each landmark count.

**Downstream grade structured $O(3)$-equivariant message-passing.** The final main experiment tests whether grade discovery can change a deeper model design. We train a small three-layer grade structured $O(3)$-equivariant message-passing network inspired by vector/bivector routing in Clifford architectures, including CGENNs (Ruhe et al., 2023a). This is not an implementation or benchmark reproduction of the published CGENN architecture. The model is deliberately small because the purpose is not to propose a new MD17 benchmark or reproduce a full CGENN benchmark suite. The architecture, optimiser, training budget, dataset and transformation distribution are fixed, and the grade assignment is the only systematic design variable. Thus the diagnostic is used as a model design step before training because it changes the in-

put routing, while the architecture and training procedure are held fixed. Here 'rule' means the deterministic target generation map used for supervised training. The target is a known $O(3)$-equivariant vector-valued rule built from force terms, scalar atom type terms and vector-bivector interactions of the form $r \times \tau$, where $\tau = r \times F$ is the torque-like axial channel and $r$ is the centred atom position. We construct this target rather than predict MD17 energies or forces because the experiment is meant to isolate parity-sensitive model selection. The rule is designed so that treating the axial channel as an ordinary vector creates a reflection law mismatch (the target is a polar vector, but the wrongly typed axial input contributes with the wrong determinant sign under reflections). An incorrect hard grade assignment therefore creates a predictable reflection law mismatch. Appendix G gives the operations and numerical equivariance checks.

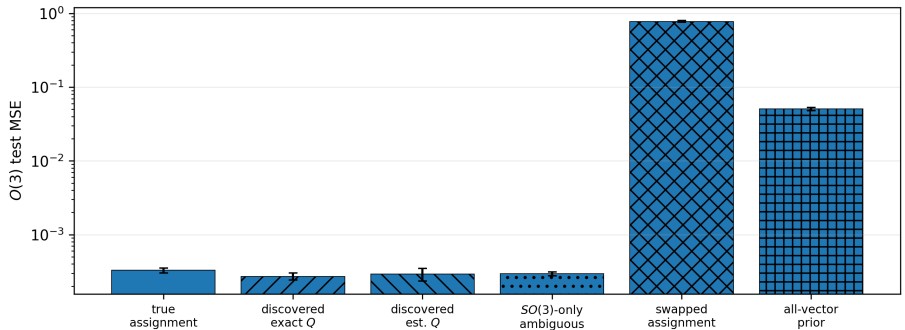

Figure 3: Downstream model selection experiment. With the same three-layer grade structured $O(3)$-equivariant architecture and training budget, the discovered assignments are close to the true assignment, while swapped and all-vector assignments suffer much larger reflection containing test error.

Table 6: Downstream $O(3)$ test MSE for the grade-structured $O(3)$-equivariant message-passing model. Values are means and standard errors over three training seeds, and all variants use the same architecture, optimiser and training distribution.

| Grade assignment | $p_{\mathrm{vec}}(F)$ | $p_{\mathrm{vec}}(r \times F)$ | $O(3)$ test MSE | Reflection-only MSE |
|---|---|---|---|---|
| true assignment | 1.000 | 0.000 | 3.29e-04 $\pm$ 2.5e-05 | 3.31e-04 $\pm$ 2.4e-05 |
| discovered exact-$Q$ assignment | 1.000 | 0.000 | 2.72e-04 $\pm$ 3.0e-05 | 2.67e-04 $\pm$ 3.8e-05 |
| discovered estimated-$Q$ assignment | 1.000 | 0.000 | 2.93e-04 $\pm$ 5.7e-05 | 2.92e-04 $\pm$ 5.7e-05 |
| $SO(3)$-only ambiguous mixture | 0.500 | 0.500 | 2.95e-04 $\pm$ 1.9e-05 | 2.94e-04 $\pm$ 1.6e-05 |
| swapped assignment | 0.000 | 1.000 | 7.78e-01 $\pm$ 2.2e-02 | 7.73e-01 $\pm$ 1.9e-02 |
| all-vector prior | 1.000 | 1.000 | 5.06e-02 $\pm$ 2.5e-03 | 5.00e-02 $\pm$ 2.7e-03 |

Figure 3 and Table 6 show that the discovered exact-$Q$ and discovered estimated-$Q$ assignments perform close to the true assignment. The swapped and all-vector hard assignments have substantially larger $O(3)$ and reflection-only test MSE than the true and discovered assignments (the values are reported in Table 6). The reflection-only evaluation in Appendix I shows the same qualitative ordering (the same ranking of assignment variants by test error), which is expected because the constructed target punishes errors in parity behaviour most directly on transformations with determinant $-1$. The $SO(3)$-only ambiguous mixture performs well on this constructed target because the architecture can still route both raw channels through both slots. In this variant, the vector input array and the bivector input array each receive a half-weighted copy of each three-coordinate channel. We therefore do not claim that ambiguity always harms downstream performance. The narrower conclusion is that when a downstream grade structured model commits to incorrect hard-coded grade choices which means a channel is placed entirely in a single vector or bivector grade rather than kept as a soft mixture, the diagnostic can identify parity errors that substantially affect reflection generalisation. The same diagnostic was also applied to the declared intermediate fields of the saved downstream models. Appendix I and Table 9 show that the vector hidden fields and bivector hidden fields are audited according to their declared laws across the recorded layers (where audited means that the same grade diagnostic is applied to paired hidden activations before and after an $O(3)$ transformation). For

every recorded layer of the true, discovered exact-$Q$ and discovered estimated-$Q$ models, declared vector hidden fields have $p_{\text{vec}} = 1.000 \pm 0.000$, while declared bivector hidden fields have $p_{\text{vec}} = 0.000 \pm 0.000$. This directly checks intermediate grade structured fields rather than only the input channels, but it does not claim arbitrary latent representation discovery without declared blocks. No hidden layer grade gate is trained in this experiment, the hidden field result is a post-hoc audit of blocks whose vector or bivector role is declared by the architecture.

Appendix I contains supporting plots for the MD17 diagnostic, the forced-$SO(3)$ negative control, reflection-only downstream evaluation, the controlled stress tests and the minimal trainable gate, together with the compact hidden layer probe table. These appendix results are used to check implementation details and failure modes. The main claims above rely only on the calibrated comparisons reported in Figures 1 to 3 and Tables 3 to 6.

## 6 Discussion and Limitations

The experiments support a calibrated claim: grade discovery audits finite, dimension-matched Clifford type assignments when paired transformed observations and a candidate representation set are available. It identifies a type only when the observed transformations separate the candidate laws, and it returns ambiguity when they do not. This is useful because it prevents rotation-only evidence from being mistaken for support of parity-sensitive grades. The diagnostic should be interpreted together with prediction loss and transformation-estimation quality. Appendix H and Table 8 provide a compact summary.

The method has a deliberately limited scope. It compares finite, dimension-matched candidate laws and therefore does not infer coordinate dimension or decompose arbitrary mixed multivectors without a known block structure. When blocks are specified, Corollary 2 extends the diagnostic to block diagonal mixed-grade assignments. The method also requires known or estimated transformations, as obtained through augmentation, controlled probing or landmark-based registration, and is not intended for unpaired observational data without such metadata. The forced-$SO(3)$ negative control in Table 4 and Appendix I shows that losing the determinant sign can make the vector/bivector diagnostic uninformative. The downstream study is a controlled model selection experiment on real molecular geometries, not a state of the art molecular prediction benchmark. Although it uses empirical positions and forces, transformed pairs are generated by controlled augmentation, and the noise study considers landmark noise in transformation estimation rather than correlated measurement noise propagated into $r \times F$. Accordingly, a near-one weight with low loss and reliable transformation metadata supports one candidate within the supplied set, and a near-half weight indicates non-identifiability and should be reported as a modelling prior unless additional parity-revealing evidence is collected. A near-one weight does not support a grade assignment when the underlying transformation metadata are unreliable.

## 7 Conclusion

We studied grade discovery as an identifiability diagnostic for Clifford-valued feature channels. The classical $SO(3)$ versus $O(3)$ parity distinction implies that rotations alone cannot separate scalar from pseudoscalar or vector from bivector in three dimensions, while reflections can. By casting this distinction as a finite candidate least-squares diagnostic, we obtain a practical test of whether a grade assignment is supported by the transformations available in a dataset or augmentation protocol. The intended role is finite candidate Clifford representation auditing rather than unrestricted representation discovery. Controlled experiments validate the theorem, MD17 real geometry experiments test physically meaningful vector and bivector channels, estimated transformation experiments quantify a realistic failure mode and the grade structured $O(3)$-equivariant message-passing experiment shows how discovered grades can alter downstream model design.

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

## A  Representation Laws and Algebraic Identities

This appendix derives the representation laws used in Eq. 1. Let $e_1, e_2, e_3$ be an orthonormal basis of $V$. The Clifford product satisfies:

$$e_i e_j + e_j e_i = 2\delta_{ij} \tag{10}$$

where $\delta_{ij}$ is the Kronecker delta. Let $Q \in O(3)$ have entries $q_{ai}$, where $i$ indexes the input basis vector and $a$ indexes the output basis coordinate. The transformed basis vector is:

$$Qe_i = \sum_{a=1}^{3} q_{ai} e_a \tag{11}$$

For a vector $v = \sum_{i=1}^{3} v_i e_i$, substitution gives:

$$Qv = Q\left(\sum_{i=1}^{3} v_i e_i\right) \tag{12}$$

$$= \sum_{i=1}^{3} v_i Q e_i \tag{13}$$

$$= \sum_{i=1}^{3} v_i \sum_{a=1}^{3} q_{ai} e_a \tag{14}$$

$$= \sum_{a=1}^{3} \left(\sum_{i=1}^{3} q_{ai} v_i\right) e_a \tag{15}$$

Thus the coordinate vector transforms as $v \mapsto Qv$. A scalar has no orientation, so $\rho_{\text{scal}}(Q) = 1$. Let $I = e_1 e_2 e_3$ be the pseudoscalar. The oriented-volume element transforms as:

$$QI = (Qe_1)(Qe_2)(Qe_3) \tag{16}$$

$$= \det(Q) e_1 e_2 e_3 \tag{17}$$

$$= \det(Q) I \tag{18}$$

Therefore $\rho_{\text{pscal}}(Q) = \det(Q)$. Finally, a bivector is represented in axial coordinates by the Hodge dual. If $b \in \mathbb{R}^3$ denotes these axial coordinates, the pseudoscalar contributes $\det(Q)$ and the axial coordinate vector rotates by $Q$, giving $b \mapsto \det(Q)Qb$. Combining these laws gives Eq. 1.

## B  Least-Squares Estimator

For two candidate laws, write $A_s$ and $B_s$ for their representation matrices at transformation $Q_s$. Let $p$ be the soft weight on $A_s$, so $1 - p$ is the weight on $B_s$. Define $D_s = A_s - B_s$. The mixed prediction for example $j$ is:

$$\widehat{y}_{sj}(p) = (pA_s + (1-p)B_s)\, x_j \tag{19}$$

$$= (B_s + p(A_s - B_s))\, x_j \tag{20}$$

$$= B_s x_j + pD_s x_j \tag{21}$$

Let $r_{sj} = y_{sj} - B_s x_j$ and $z_{sj} = D_s x_j$. The loss is:

$$L(p) = \frac{1}{SN} \sum_{s=1}^{S} \sum_{j=1}^{N} \|r_{sj} - pz_{sj}\|_2^2 \tag{22}$$

$$= \frac{1}{SN} \sum_{s,j} \left(r_{sj}^\top r_{sj} - 2p r_{sj}^\top z_{sj} + p^2 z_{sj}^\top z_{sj}\right) \tag{23}$$

Taking the derivative with respect to $p$ and setting it to zero gives:

$$p = \frac{\sum_{s,j} r_{sj}^\top z_{sj}}{\sum_{s,j} z_{sj}^\top z_{sj}} \tag{24}$$

The implementation clips this value to $[0, 1]$. If the denominator is zero, the candidate predictions are identical on the observed examples and the diagnostic returns $p = 1/2$.

## C  Proof of Finite Candidate Identifiability

Define the population residual matrix:

$$M_p(Q) = \rho_{t_\star}(Q) - \sum_{t \in \mathcal{C}_d} p_t \rho_t(Q) \tag{25}$$

Since $y = \rho_{t_\star}(Q)x$, the population loss is:

$$\mathcal{L}(p) = \mathbb{E}\left[\left\|\rho_{t_\star}(Q)x - \sum_{t \in \mathcal{C}_d} p_t \rho_t(Q)x\right\|_2^2\right] \tag{26}$$

$$= \mathbb{E}\left[\|M_p(Q)x\|_2^2\right] \tag{27}$$

Conditioning on $Q$ gives:

$$\mathbb{E}\left[\|M_p(Q)x\|_2^2 \mid Q\right] = \mathrm{tr}\left(M_p(Q)^\top M_p(Q)G(Q)\right) \tag{28}$$

Because $G(Q)$ is positive definite almost surely, this trace is zero if and only if $M_p(Q) = 0$. Hence $\mathcal{L}(p) = 0$ if and only if the mixed representation equals the true representation almost surely. This is why the theorem uses an almost sure condition rather than requiring equality for transformations that have zero probability under the observed protocol. The separation condition in Eq. 5 states that only the one-hot vector on $t_\star$ has this property, which proves uniqueness. Conversely, if two candidate laws are equal almost surely, transferring probability mass between those candidates leaves the mixed representation unchanged and the loss cannot distinguish them.

## D  Proof of the Three-Dimensional Parity Proposition

For rotations, $\det(Q) = 1$, so Eq. 1 gives:

$$\rho_{\mathrm{scal}}(Q) = \rho_{\mathrm{pscal}}(Q) = 1, \qquad \rho_{\mathrm{vec}}(Q) = \rho_{\mathrm{biv}}(Q) = Q \tag{29}$$

Thus rotation-only evidence makes each dimension-matched pair identical for every observed sample and Theorem 1 gives non-identifiability.

For a reflection, $\det(Q) = -1$, so:

$$\rho_{\mathrm{pscal}}(Q) = -\rho_{\mathrm{scal}}(Q), \qquad \rho_{\mathrm{biv}}(Q) = -\rho_{\mathrm{vec}}(Q) \tag{30}$$

Consider the vector/bivector pair; the scalar/pseudoscalar pair is the one-dimensional analogue. If the true-type is vector and the mixture puts weight $p$ on the vector law, the predicted matrix on a reflection is $(2p-1)Q$. Equality with the true vector law $Q$ requires $p = 1$. If the true-type is bivector, equality with $-Q$ requires $p = 0$. Since $G(Q)$ is positive definite almost surely, these matrix differences are visible to the loss. Any positive reflection probability therefore identifies the true member of the pair in the population setting.

# E Finite-Sample Probability of Observing Parity Evidence

Let $r$ be the probability that one sampled transformation is a reflection. The probability that this transformation is not a reflection is $1 - r$. Since $S$ transformations are sampled independently, the probability that none is a reflection is:

$$P(\text{no reflection}) = (1 - r)^S \tag{31}$$

Consequently, the probability of at least one reflection is:

$$P_{\text{sep}}(r, S) = 1 - (1 - r)^S \tag{32}$$

If no reflection gives reported true-type weight $1/2$ and at least one reflection gives weight 1, then:

$$W_{\text{true}}(r, S) = \frac{1}{2}P(\text{no reflection}) + P(\text{at least one reflection}) \tag{33}$$

$$= \frac{1}{2}(1 - r)^S + 1 - (1 - r)^S \tag{34}$$

$$= 1 - \frac{1}{2}(1 - r)^S \tag{35}$$

The same finite-sample probability applies to the vector/bivector pair. Appendix D shows that a reflection separates the vector and bivector matrices through the same sign flip that separates scalar and pseudoscalar laws. Thus, one observed reflection supplies parity separating evidence for either pair, while in the population setting any positive reflection probability identifies either pair under the conditional second moment condition.

# F Estimated Transformations and Experimental Details

For the MD17 and downstream experiments, row vector coordinates are transformed as $P' = PQ^\top$, which is equivalent to the main text column vector representation convention. The orthogonal matrices $Q$ are sampled exactly as in the controlled benchmark. For rotations, we draw four independent standard normal numbers, normalise the resulting quaternion to unit length and convert that unit quaternion to a rotation matrix. Reflection examples compose the rotation with $\text{diag}(-1, 1, 1)$. To estimate $Q$, we centre the source landmarks $P$ and transformed landmarks $P'$, form the cross-covariance matrix and apply the Kabsch orthogonal Procrustes estimator. The computation uses a singular value decomposition (SVD). In the $O(3)$-valued setting, the SVD solution is not constrained to be a rotation and can have determinant $-1$. In the forced-$SO(3)$ negative control, the last singular vector is flipped when necessary so that $\det(\widehat{Q}) = 1$. Landmark noise is added to transformed landmarks before estimation and is scaled by the empirical standard deviation of the transformed landmark coordinates. For subset landmark experiments with $K \in \{3, 4, 6\}$, atoms are sampled uniformly without replacement independently for each diagnostic pair. The setting $K = 9$ uses all ethanol atoms. This means the $K = 3$ case is not hand picked to be favourable and explains why it often fails to recover handedness. The purpose of this registration stage is only to obtain the transformation metadata that the diagnostic consumes. We use Kabsch alignment as the standard SVD-based baseline because it returns the orthogonal matrix $\widehat{Q}$ used directly in the diagnostic. Geometric algebra registration methods address the same metadata problem with rotors instead of matrices. Examples include rotor-based solutions to three-dimensional registration and absolute orientation (Matsantonis & Lasenby, 2023; 2024; Matsantonis, 2025). These are natural alternatives for a future transformation estimation stage, but they are not used in our experiments. Our reported estimated-$Q$ results therefore reflect the Kabsch baseline rather than a comparison among registration algorithms.

The reported run loaded PyTorch Geometric's predefined MD17 ethanol CCSD(T) training split and selected 700 frames without replacement using seed 12345. The first 500 selected frames were used for training and the remaining 200 as an internal held-out evaluation set. Scalar normalization constants for positions, forces, $r \times F$ and the constructed target were computed over the 700 selected frames before this split. The run used 64 diagnostic pairs per seed, eight grade-discovery seeds, landmark counts $K \in \{3, 4, 6, 9\}$, landmark noise values $\{0, 0.005, 0.01, 0.02, 0.05, 0.10, 0.20, 0.35, 0.50, 0.75, 1.0\}$, three downstream training seeds, 450 downstream optimisation steps, hidden width 12, three message-passing layers, AdamW with learning rate 0.003 and weight decay $10^{-5}$, batch size 32, 24 evaluation batches and radial scale 1.25.

## G   Downstream Model and Implementation Details

The downstream experiment uses a small grade-structured $O(3)$-equivariant message-passing network inspired by Clifford architectures. The input contains standardised atomic number scalar features, a polar vector force channel, and an axial vector channel $\tau_i = r_i \times F_i$. A grade assignment routes each raw three-dimensional channel into vector slots with weight $p_{\text{vec}}$ and into bivector slots with weight $1 - p_{\text{vec}}$. The true assignment uses $p_{\text{vec}}(F) = 1$ and $p_{\text{vec}}(\tau) = 0$. The discovered assignments use the weights fitted by the diagnostic, while the swapped and all-vector baselines use deliberately incorrect or naive routing.

Each message-passing layer uses scalar radial weights that depend only on pairwise distances. Its vector updates combine existing vectors, scalar multiples of relative unit vectors, and cross products between relative vectors and axial bivectors. Its bivector updates combine existing bivectors and cross products between polar vectors. These operations preserve the intended $O(3)$ laws because scalar radial weights are invariant, polar vectors transform by $Q$, axial vectors transform by $\det(Q)Q$ and the cross product contributes the determinant factor required by the parity of the output. All grade assignment variants use the same architecture, optimiser and training distribution.

The vector target is generated from a known $O(3)$-equivariant rule. For atom $i$, neighbour $j$, relative unit vector $\hat{r}_{ij}$, radial weight $w_{ij}$, scalar atom feature $a_j$, force $F_j$, and axial channel $\tau_j$, the unnormalised target has the form:

$$y_i = \sum_{j \neq i} w_{ij} \left(F_j + 0.70\,\hat{r}_{ij} \times \tau_j + 0.15\,a_j\hat{r}_{ij}\right) + 0.20F_i + 0.10\,r_i \times \tau_i \tag{36}$$

The constants are fixed and are not tuned per variant. This target is not meant to be a new molecular benchmark. It is an auditable parity-sensitive task on real molecular geometries, chosen specifically so that the parity of the axial channel affects the supervised vector output under reflections.

Table 7 reports numerical equivariance checks for the target rule, the model output, and the hidden fields. The relative errors are small, with model and hidden field checks below approximately $3 \times 10^{-4}$ in relative root mean squared error (RMSE) and the target rule near $10^{-7}$. These checks verify that the implementation follows the declared transformation laws to the numerical precision relevant for the experiments.

Table 7: Numerical equivariance sanity checks for the downstream experiment.

| Check | Relative RMSE | Max absolute error |
|---|---|---|
| hidden bivector law | 2.97e-04 $\pm$ 3.4e-06 | 6.88e-04 $\pm$ 5.4e-05 |
| hidden scalar invariant | 1.66e-05 $\pm$ 3.8e-06 | 1.05e-04 $\pm$ 2.4e-05 |
| hidden vector law | 2.91e-04 $\pm$ 1.7e-06 | 9.76e-04 $\pm$ 1.4e-04 |
| model output vector | 1.83e-04 $\pm$ 2.5e-05 | 4.41e-04 $\pm$ 2.2e-05 |
| target rule vector | 2.83e-07 $\pm$ 1.9e-08 | 1.03e-06 $\pm$ 2.1e-07 |

Figure 4 visualises the same checks as Table 7. We include the figure as an implementation diagnostic rather than as an additional performance claim. The target rule vector error is near machine precision, while the model output and hidden-field errors remain below the scale relevant for the reported downstream MSE comparisons.

## H   Diagnostic Interpretation Summary

Table 8 summarises how we interpret the diagnostic outputs used throughout the paper. It is not a universal thresholding rule. It records the qualitative reading of soft weights, prediction loss and transformation estimation reliability within the finite candidate set supplied by the practitioner.

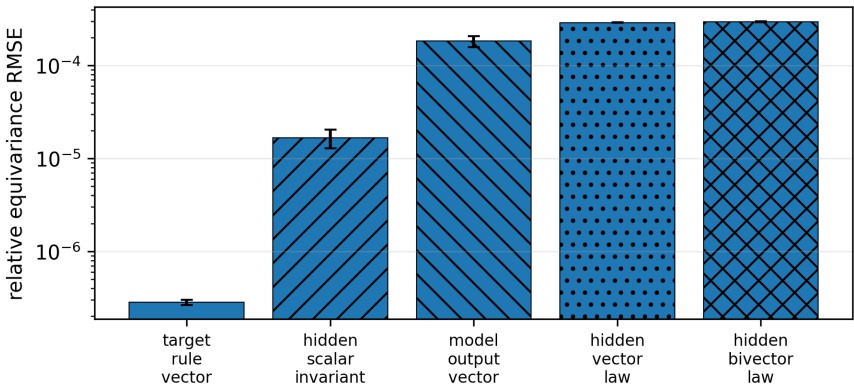

Figure 4: Equivariance sanity checks for the downstream target, model output and hidden fields. The plot is a visual companion to Table 7.

Table 8: Interpreting the diagnostic output.

| Observed diagnostic behaviour | Interpretation | Recommended use |
|---|---|---|
| Near-one weight with low prediction loss and reliable transformation metadata | The supplied evidence supports one candidate within the finite set. | Use the candidate as a data supported grade assignment, subject to the declared scope. |
| Near-half weight for a two-candidate parity pair | The observed transformations do not identify the grade. | Report the grade as a modelling prior or collect parity revealing evidence. |
| Near-one weight with poor prediction loss or unreliable determinant signs | The transformation law or transformation estimate may be misspecified. | Treat the result as a warning and inspect the registration or candidate set. |

## I    Additional Experimental Results

Figure 5 gives the full real geometry diagnostic plot corresponding to Table 4. It visualises that the exact-$O(3)$ and estimated-$Q$ settings separate force from $r \times F$, while the rotation-only and forced-$SO(3)$ settings do not. Figure 6 isolates the forced-$SO(3)$ negative control for estimated transformations and shows that losing determinant sign information returns the diagnostic to ambiguity. Table 9 reports the hidden layer probe numerically rather than as an additional figure, because the table is the more compact way to record the declared vector and bivector hidden field checks. The hidden probe is a check of the declared grade structured implementation, not evidence that arbitrary latent features can be decomposed without a candidate structure.

Figure 7 shows the controlled scalar/pseudoscalar loss landscape corresponding to the main-text identifiability discussion. Figure 8 retains the controlled stress analyses. Its left panel shows that approximate but determinant-preserving orthogonal metadata increases prediction error, while its right panel shows that low variance in the separating coordinate reduces the evidence denominator. Figure 9 gives the reflection-only downstream evaluation, which is included because parity errors are most visible on transformations with determinant $-1$. Table 10 records the controlled single-run recovery values used to check the noise floor and denominator behaviour. Table 11 reports the minimal trainable gate. It is retained only to show that the softmax gate can be optimised with another parameter, and it is not used as evidence for full downstream Clifford network performance.

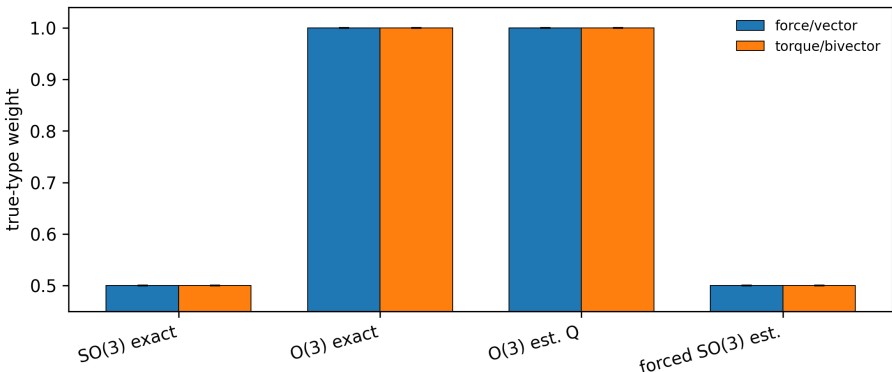

Figure 5: Real geometry grade discovery on MD17 ethanol channels. Rotations alone remain ambiguous, while exact and estimated $O(3)$ evidence recover force as a vector and torque-like cross products as bivectors.

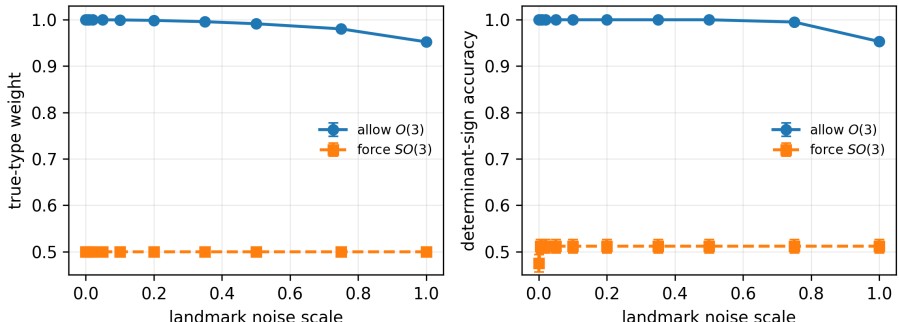

Figure 6: Forced-$SO(3)$ transformation estimation is a negative control. When reflected pairs are forced into a rotation-only estimate, determinant sign information is lost and the grade diagnostic becomes uninformative.

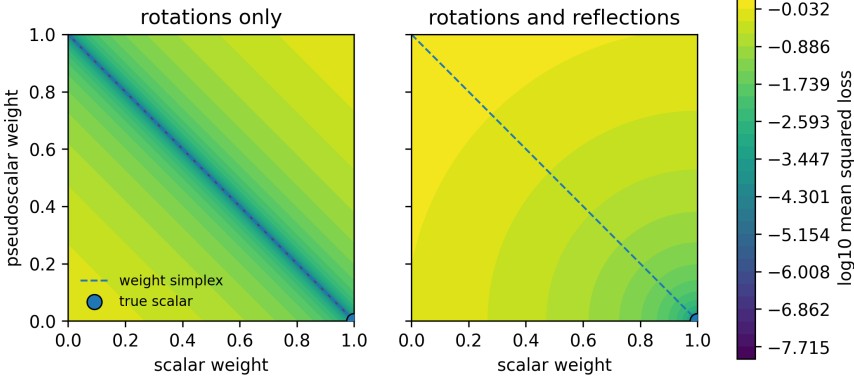

Figure 7: Controlled scalar-pseudoscalar loss landscape. Rotation-only evidence creates a flat valley on the weight simplex, while reflections create a unique minimum at the true scalar law. This visualisation is a geometric counterpart of Theorem 1, not a separate empirical claim.

Table 9: Compact hidden layer probe summary. The table reports vector law weights for declared vector and bivector hidden fields across saved downstream models.

| Model | Layer | Field | $p_{vec}$ | True-type weight |
|---|---|---|---|---|
| O3 train discovered O3 | 0 | bivector hidden | $0.000 \pm 0.000$ | $1.000 \pm 0.000$ |
| O3 train discovered O3 | 1 | bivector hidden | $0.000 \pm 0.000$ | $1.000 \pm 0.000$ |
| O3 train discovered O3 | 2 | bivector hidden | $0.000 \pm 0.000$ | $1.000 \pm 0.000$ |
| O3 train discovered O3 | 3 | bivector hidden | $0.000 \pm 0.000$ | $1.000 \pm 0.000$ |
| O3 train discovered O3 | 0 | vector hidden | $1.000 \pm 0.000$ | $1.000 \pm 0.000$ |
| O3 train discovered O3 | 1 | vector hidden | $1.000 \pm 0.000$ | $1.000 \pm 0.000$ |
| O3 train discovered O3 | 2 | vector hidden | $1.000 \pm 0.000$ | $1.000 \pm 0.000$ |
| O3 train discovered O3 | 3 | vector hidden | $1.000 \pm 0.000$ | $1.000 \pm 0.000$ |
| O3 train discovered estimated Q | 0 | bivector hidden | $0.000 \pm 0.000$ | $1.000 \pm 0.000$ |
| O3 train discovered estimated Q | 1 | bivector hidden | $0.000 \pm 0.000$ | $1.000 \pm 0.000$ |
| O3 train discovered estimated Q | 2 | bivector hidden | $0.000 \pm 0.000$ | $1.000 \pm 0.000$ |
| O3 train discovered estimated Q | 3 | bivector hidden | $0.000 \pm 0.000$ | $1.000 \pm 0.000$ |
| O3 train discovered estimated Q | 0 | vector hidden | $1.000 \pm 0.000$ | $1.000 \pm 0.000$ |
| O3 train discovered estimated Q | 1 | vector hidden | $1.000 \pm 0.000$ | $1.000 \pm 0.000$ |
| O3 train discovered estimated Q | 2 | vector hidden | $1.000 \pm 0.000$ | $1.000 \pm 0.000$ |
| O3 train discovered estimated Q | 3 | vector hidden | $1.000 \pm 0.000$ | $1.000 \pm 0.000$ |
| O3 train true assignment | 0 | bivector hidden | $0.000 \pm 0.000$ | $1.000 \pm 0.000$ |
| O3 train true assignment | 1 | bivector hidden | $0.000 \pm 0.000$ | $1.000 \pm 0.000$ |
| O3 train true assignment | 2 | bivector hidden | $0.000 \pm 0.000$ | $1.000 \pm 0.000$ |
| O3 train true assignment | 3 | bivector hidden | $0.000 \pm 0.000$ | $1.000 \pm 0.000$ |
| O3 train true assignment | 0 | vector hidden | $1.000 \pm 0.000$ | $1.000 \pm 0.000$ |
| O3 train true assignment | 1 | vector hidden | $1.000 \pm 0.000$ | $1.000 \pm 0.000$ |
| O3 train true assignment | 2 | vector hidden | $1.000 \pm 0.000$ | $1.000 \pm 0.000$ |
| O3 train true assignment | 3 | vector hidden | $1.000 \pm 0.000$ | $1.000 \pm 0.000$ |

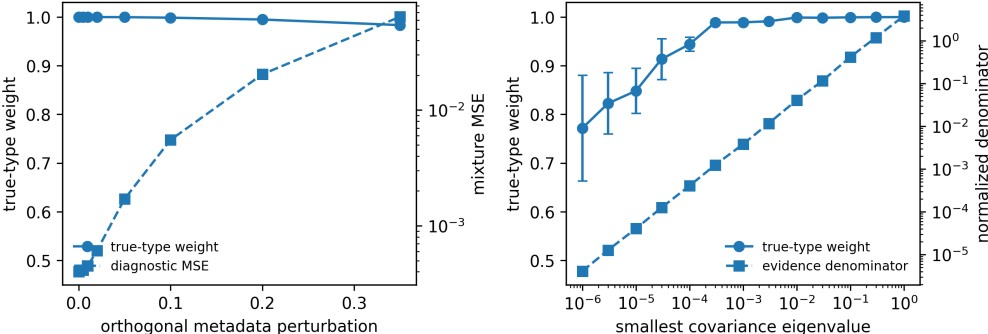

Figure 8: Controlled stress tests. Approximate orthogonal metadata increases prediction error and low covariance in the separating direction reduces the evidence denominator.

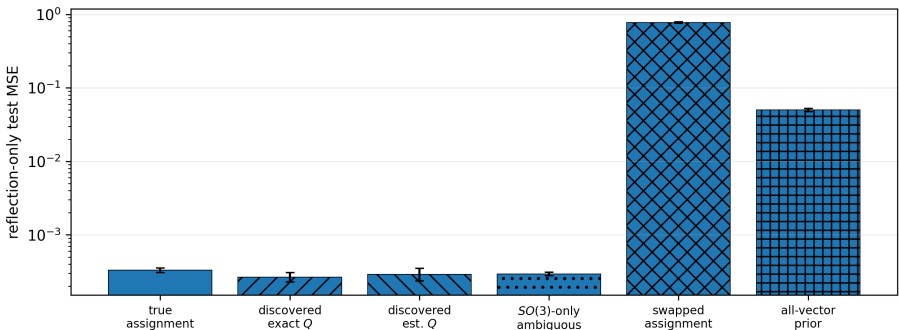

Figure 9: Reflection-only downstream evaluation. The qualitative conclusion matches Table 6. True and discovered assignments remain close, while incorrect hard assignments have much larger error.

Table 10: Single run controlled recovery table from the controlled benchmark. The zero evidence denominator under rotations explains why the diagnostic reports weight 1/2, whereas the positive denominator under reflections corresponds to identifiable parity evidence.

| Transformation family | True type | True-type weight | Mixture MSE | Evidence denominator |
|---|---|---|---|---|
| rotations | scalar | 0.500 | 0.0004 | 0.0 |
| rotations | pseudoscalar | 0.500 | 0.0004 | 0.0 |
| rotations | vector | 0.500 | 0.0004 | 0.0 |
| rotations | bivector | 0.500 | 0.0004 | 0.0 |
| rotations + reflections | scalar | 1.000 | 0.0004 | 6825.6 |
| rotations + reflections | pseudoscalar | 1.000 | 0.0004 | 4768.7 |
| rotations + reflections | vector | 1.000 | 0.0004 | 30944.5 |
| rotations + reflections | bivector | 1.000 | 0.0004 | 17221.8 |

Table 11: Minimal trainable gate. This experiment is retained only as an optimisation check. It verifies that the soft grade gate can be trained together with a learned scalar gain, while the downstream experiment in the main text is the evidence for deeper grade structured model selection.

| Train transformations | Test transformations | Final true-type weight | Learned $\gamma$ | Test MSE | Seeds |
|---|---|---|---|---|---|
| rotations only | rotations only | $0.500 \pm 0.000$ | $1.000 \pm 0.000$ | $1.18\text{e-}07 \pm 2.2\text{e-}08$ | 12 |
| rotations only | rotations + reflections | $0.500 \pm 0.000$ | $1.000 \pm 0.000$ | $4.95\text{e-}01 \pm 1.3\text{e-}02$ | 12 |
| rotations + reflections | rotations + reflections | $0.930 \pm 0.001$ | $1.068 \pm 0.003$ | $5.95\text{e-}03 \pm 1.4\text{e-}04$ | 12 |

