# OpenReview forum: "Grade Discovery as an Identifiability Diagnostic for Clifford-Valued Features"
_TMLR — Under review for TMLR_

### Review · Reviewer_cHsZ · 2026-06-22

**Summary Of Contributions:**

The paper introduces grade discovery as an identifiability diagnostic to determine the geometric type of Clifford-valued feature channels. In geometric machine learning, feature types are typically predefined by the practitioner before training. This paper challenges this by analyzing when these types are mathematically identifiable from data.

In detail, the paper formulates this as a dimension-matched type estimation problem optimized through a differentiable equivariant loss. Moreover, it provides an identifiability theorem proving that rotation-only evidence (SO(3)) cannot distinguish between scalar/pseudoscalar or vector/bivector pairs in three dimensions, while the introduction of reflections (O(3)) resolves this ambiguity. And it implements abundant experiments to empirically validate the theoretical conclusions.

Strengths:
1. This paper has demonstrated a clear framework of theory by treating geometric type assignment as an auditable, data-driven identifiability problem.

2. The math derivatives and theoretical analysis are rigorous, with detailed discussions and proofs.

3. This paper provides comprehensive experiments to validate the proposed diagonostic for checking whether a Clifford valued feature assignment is supported by transformation evidence.

Weakness:

1. The diagnostic relies on paired observations (before and after some known transformations Q). Although the authors note that these are easily synthesized via data augmentation, this requirement limits its direct application to raw, observational datasets where transformations are unknown.

2. The theoretical finding that scalars/pseudoscalars and vectors/bivectors are indistinguishable under proper rotations but distinguishable under reflections is a basic consequence of classical group representation theory. Therefore, the theoretical results have limited novelty.

3. The empirical evaluation of the differentiable gate is restricted to a minimal single-layer prediction module instead of a real, multi-layer Clifford Neural Network. While the paper explains that it is designed to keep the diagnostic isolated, it remains unknown for the practicability of the proposed method in deeper networks.

4. The framework assumes that the coordinate dimension is already known (e.g., only comparing 3D vectors to 3D bivectors). In many practical geometric algebra applications, the primary difficulty lies in choosing the correct embedding dimension. By restricting the candidates to the same dimension, the paper bypasses the more challenging and practical aspect of type discovery.

**Audience:**

Yes

**Audience Explanation:**

Researchers working on geometric deep learning, equivariant neural networks, and Clifford/Geometric Algebra-based architectures would find this work highly relevant. For instance, it reveals that standard rotation-only data augmentation is fundamentally insufficient if a model genuinely relies on isolating proper parity characteristics. The insights regarding representation separation and the necessity of specific transformation families are valuable for model design, data augmentation strategies, and debugging equivariance.

**Broader Impact Concerns:**

There are no broader impact concerns for this submission. The work is a foundational, mathematical study of representation identifiability in geometric machine learning. It does not involve human subjects, sensitive datasets, or other issues that requires adding a Broader Impact Statement.

**Claims And Evidence:**

Yes

**Claims Explanation:**

The claims made in the manuscript are supported mostly by technically sound and clear evidence except some limitations.

- Claim 1: Rotations alone cannot identify parity-sensitive pairs in 3D. This is proven analytically in Proposition 1 and demonstrated empirically in Table 1 and Figure 1.

- Claim 2: Reflections resolve the ambiguity. Supported by Table 1, where the true-type weight reaches 1.0 when reflections are introduced, and by Figure 2, which visually demonstrates how reflections break the flat valley in the loss landscape to create a unique minimum.

- Claim 3: Recovery rates follow the analytical probability curve. The right panel of Figure 1 shows that the empirical simulation points closely track the noiseless mathematical calculation in (8).

- Claim 4: Sensitivity to noise and degeneracy. The stress tests in Figure 4 demonstrate the degradation of the diagnostic under inaccurate orthogonal matrices and low-variance coordinates, confirming that the assumptions of Theorem 1 are indeed operational requirements. However, in Figure 4, even moderate perturbation of Q degrades the diagnostic's accuracy. Actually, in real-world data, transformations are rarely clean orthogonal matrices, which significantly limits the claims of practical applicability.

**Requested Changes:**

1. The current paper isolates the diagnostic to a single-layer setup to "remove optimization noise". However, to prove that this diagnostic is practically useful, it is better to demonstrate its performance in a deeper network architecture. Please show whether the soft-gating mechanism can successfully identify the correct grades of intermediate layers on deeper models.

2. In practical settings, the exact orthogonal transformation Q is rarely known. Modify the noise stress test to evaluate the performance of the diagnostic when Q must be estimated from noisy landmarks or paired points rather than being given. Please show the breakdown threshold of the diagnostic under estimated transformations.

3.  Revise the introduction and conclusion to clearly acknowledge that the indistinguishability of these representations is a classical algebraic fact under SO(3) v.s. O(3). The paper's contribution is an empirical verification and optimization-based formulation of this known representation theory phenomenon within machine learning instead of a completely new theoretical discovery.

---

> ### Author Response · Authors · 2026-07-10
> **Response to Reviewer cHsZ**
>
> We thank the reviewer for the positive and constructive review. We are grateful that the reviewer found the theoretical framework and experiments clear. The requested changes helped us substantially improve the practical evidence, the treatment of estimated transformations and the positioning of the contribution relative to classical representation theory.
>
> Requested Change --> Demonstrate the diagnostic in a deeper network architecture and show whether the soft-gating mechanism can identify correct grades of intermediate layers.
>
> We added a controlled three-layer grade-structured O(3)-equivariant message-passing experiment in Section 5 (Figure 3 and Table 6). The diagnostic is run before network training, and its estimated weights are then fixed as routing coefficients for the force and torque-like input channels. The experiment therefore tests whether grade discovery changes the design and reflection generalization of a deeper equivariant model. It does not train a separate soft grade gate at every hidden layer. We additionally apply the diagnostic post hoc to the architecture’s declared vector and bivector hidden blocks. Appendix I, Table 9 reports p_vec=1.000+-0.000 for declared vector fields and p_vec=0.000+-0.000 for declared bivector fields across the recorded layers of the true and discovered models. This is an implementation audit of declared hidden fields, not arbitrary hidden layer grade discovery. Appendix G gives the architecture and equivariance checks.
>
> Requested Change --> Evaluate the diagnostic when Q is estimated from noisy landmarks or paired points, and show a breakdown threshold.
>
> We added a new estimated transformation experiment in Section 5. In the MD17 ethanol setting, Q is estimated from paired atom positions using Kabsch/orthogonal Procrustes alignment after adding Gaussian landmark noise. Figure 2 reports the true-type weight, determinant sign accuracy, and Frobenius error as landmark noise and landmark count vary. Table 5 reports the operational breakdown threshold, defined as the first tested landmark noise level at which the average true-type weight falls below 0.90. Appendix F gives the detailed Kabsch procedure, the landmark noise scaling, the landmark counts K in {3,4,6,9}  and the transformation sampling distribution. We also include a forced-SO(3) negative control in Section 5/Table 4 and Appendix I/Figure 6 showing that when determinant sign information is lost the diagnostic returns to ambiguity.
>
> Requested Change --> Acknowledge that the SO(3) vs. O(3) indistinguishability is a classical algebraic fact, and frame the contribution as an ML diagnostic built from it.
>
> We revised the Abstract, Introduction, controlled experiment discussion and Conclusion accordingly. The Abstract now explicitly states that the classical representation theoretic fact behind the problem is that rotations cannot distinguish scalar from pseudoscalar or vector from bivector in three dimensions, whereas reflections can, and that our contribution is an auditable machine learning test built from this fact. In Section 5, the controlled experiments are described as sanity checks that the diagnostic reports ambiguity exactly when the candidate laws are indistinguishable. The Conclusion now states that the intended role is finite candidate Clifford representation auditing rather than unrestricted representation discovery or a new algebraic theorem.
>
> Additional clarification on coordinate dimension.
>
> Although this was listed as a limitation rather than a requested change, we also clarified the scope in Sections 3 and 6. The method intentionally assumes a declared coordinate block and does not solve embedding dimension discovery. We added Assumption 1 ('Blockwise homogeneous candidates') and Corollary 2 ('Blockwise finite candidate products') to show how the same finite
> candidate logic applies to block diagonal mixed-grade assignments when the block partition is known.
>
> We hope these revisions address the concerns while preserving the calibrated scope of the diagnostic.

---

### Review · Reviewer_RBSg · 2026-06-24

**Summary Of Contributions:**

This paper studies grade discovery for Clifford valued features. The central problem is: given paired observations before and after known geometric transformations, can one infer whether a feature channel should be treated as a specific grade of Clifford algebra? The proposed method restricts the problem to dimension-matched candidate grades, for example scalar vs. pseudoscalar for one-dimensional channels and vector vs. bivector for three-dimensional channels. It then fits a soft mixture over predefined candidate representation matrices using a least-squares equivariant prediction loss. The paper proves an identifiability result: a grade can be recovered when the observed transformations separate the candidate representation laws. In the main three-dimensional setting, rotations alone cannot distinguish scalar from pseudoscalar or vector from bivector, while reflections can separate these pairs under nondegenerate data covariance. The experiments on synthetic tests involving rotations and reflections support the conclusion.

**Audience:**

No

**Audience Explanation:**

I do not think the findings are likely to be broadly interesting to the TMLR audience. The main result is a narrow identifiability diagnostic for predefined, dimension-matched Clifford grade candidates under known transformations. It does not discover general representations, infer unknown symmetries, guide the design of a equivariant neural network, or demonstrate improved downstream ML performance. Thus, although technically coherent, the contribution appears too narrow and specialized for TMLR.

**Broader Impact Concerns:**

I do not see ethical or societal risks specific to this work.

**Claims And Evidence:**

Yes

**Claims Explanation:**

The paper’s narrow mathematical claim is largely correct and clearly demonstrated: if two candidate representation laws are identical on the observed transformations, the least-squares loss cannot distinguish them; if reflections are present, scalar vs. pseudoscalar and vector vs. bivector become distinguishable in the O(3) setting.

However, the contribution is too narrow to provide useful guidance for equivariant neural network design. The estimator assumes paired observations, known transformation matrices and dimension-matched candidate sets. This is a very restricted selection problem.

**Requested Changes:**

- **Demonstrate practical value for machine learning.** The current experiments do not show that grade discovery helps select equivariant features or design a equivariant neural network. A convincing revision should include at least one nontrivial Clifford Group Equivariant neural network [A] experiment where the detection of grade changes the network design and improves downstream performance.

- **Add realistic data settings.** The current benchmark is synthetic and generated exactly from the assumed laws. The paper should show how often the assumed paired observations and known transformations actually arise in practical machine learning tasks, and should test the method under more realistic conditions. Add clearer guidance on how authors can obtain paired observations and known transformation matrices in real datasets.

- **Substantially improve the related work and discussion.** The paper should discuss the broader literature on symmetry discovery [B, C] and group representation learning [D]. The authors should clearly explain whether their contribution is merely a finite-candidate detection specific to representation of Clifford algebra or whether it offers something conceptually new beyond existing representation discovery methods.

- **Expand the formulation beyond very limited candidate pairs.** The current setup handles only scalar vs. pseudoscalar and vector vs. bivector in three dimensions for O(3) group. It does not decide whether a channel should be one-dimensional or three-dimensional, nor does it discover unknown representation laws. A stronger paper would extend the method to larger candidate sets, mixed-grade multivectors or more general transformation groups.

[A] Ruhe et al. "Clifford group equivariant neural networks." NeurIPS 2023.

[B] Yang et al. "Generative adversarial symmetry discovery." ICML 2023.

[C] Desai et al. "Symmetry discovery with deep learning." Physical Review D 105.9 (2022): 096031.

[D] Quessard et al. "Learning disentangled representations and group structure of dynamical environments." NeurIPS 2020.

---

> ### Author Response · Authors · 2026-07-10
> **Response to Reviewer RBSg**
>
> We thank the reviewer for the thoughtful review. We took the concern about narrowness seriously. In the revision we added: (i) a real geometry MD17 diagnostic, (ii) an estimated transformation experiment from noisy landmarks, (iii) a deeper grade structured O(3)-equivariant message-passing model inspired by Clifford architectures in which the discovered grades change input routing, (iv) an expanded related work/positioning discussion and (v) a blockwise extension for finite mixed-grade candidate sets. We also clarified that the method is not general symmetry or representation discovery.
>
> Requested Change --> Demonstrate practical value for machine learning, including an experiment where grade detection changes equivariant-network design.
>
> We added a new downstream experiment in Section 5 using a small three-layer grade structured O(3)-equivariant message-passing network inspired by vector/bivector routing in Clifford architectures including CGENNs [Ruhe et al., 2023]. It is not a reproduction of the published CGENN architecture or a SOTA MD17 benchmark. Instead, it is a controlled model selection test, i.e. the architecture, optimizer, training budget, data, target rule and transformation distribution are fixed, and the grade assignment is the only systematic design variable. Figure 3 and Table 6 show that the discovered exact-Q and estimated-Q assignments perform close to the true assignment, while swapped and all-vector hard assignments have much larger O(3) test MSE. Appendix G gives the model operations and equivariance checks, and Appendix I gives the reflection-only evaluation and hidden field audit.
>
> Requested Change --> Add realistic data settings and clarify how paired observations and transformations arise.
>
> We added real geometry experiments on MD17 ethanol in Section 5 ('Real geometry diagnostic'). The diagnostic is applied to non-Gaussian molecular channels, i.e. force F_i treated as a polar vector and tau_i= r_i x F_i treated as an axial vector/bivector. Table 4 reports that exact O(3) evidence recovers these channels, while rotation-only evidence remains ambiguous. We also added in Section 5 ('Estimated transformations from landmarks'), where Q is estimated by Kabsch alignment from noisy paired atom positions. Figure 2 and Table 5 report degradation and operational breakdown thresholds as landmark noise and landmark count vary. Appendix F specifies the transformation distribution and Kabsch details. The Discussion now states that paired evidence can arise from augmentation, physical/simulation probing or registration, and that the diagnostic is not intended for unpaired observational data without such metadata. We also clarify that the transformed feature pairs are generated by controlled augmentation and that correlated measurement noise propagated into r x F is not modeled.
>
> Requested Change --> Substantially improve related work and clarify relation to symmetry discovery and representation learning.
>
> We expanded Section 2 with a new 'Symmetry and representation discovery' discussion covering LieGAN, SymmetryGAN and Quessard et al. We also added Table 2 in Section 2. The revision explicitly states that grade discovery is finitecandidate Clifford representation auditing, not unrestricted representation discovery. The key distinction is that even if a symmetry-discovery method recovers an SO(3) action, this does not distinguish polar-vector from axial-bivector laws, because they coincide on SO(3). Thus our diagnostic asks a downstream identifiability question, i.e. once a transformation family or estimated matrix is available, does it separate the Clifford candidate laws used by the model?
>
> Requested Change --> Expand beyond very limited candidate pairs.
>
> We clarified the scope and added a formal extension. Section 3 includes Assumption 1 ('Blockwise homogeneous candidates'), and Section 4 adds Corollary 2 ('Blockwise finite candidate products') extending Theorem 1 to block diagonal finite candidate assignments for multivectors with known block partitions. This is a formal extension only, i.e. the revision does not add an empirical larger candidate benchmark, infer the block partition, decide the coordinate dimension or discover unknown representation laws. Section 6 states these boundaries explicitly.
>
> We hope these additions make the paper more useful to TMLR readers interested in equivariant model design and clarify the intended, deliberately scoped role of the diagnostic.

---

### Review · Reviewer_ErtS · 2026-06-28

**Summary Of Contributions:**

It is identified that most Clifford-algebra based approaches to learning hand-specify the grade(s) of the inputs to their respective models, which essentially comes from domain knowledge. To that end, the authors attempt to identify when such information isn't available and whether, in those cases, they can manually 'discover' the grade of each channel. Focusing on Cl_3 multivectors and two types of channels (1 and 3 dimensional) their primary objective is to separate scalar v pseudoscalar for 1 dimensional and vector v bivector for 3 dimensional cases. The problem is simplified by focusing on pairs of observations before and after 'known' transformations from O(3). It is noted that only under reflections the representation matrices can actually differentiate between the two grades in each case, and subsequently a logical approach based on soft-weights and linear models is proposed to identify the grade type based on how the data transforms. Experiments find that rotations cannot reveal grade type as expected, and reflections can. Further experiments target different stress tests based on noisy observations on the data side or the matrix side, among others.

**Audience:**

Yes

**Audience Explanation:**

One of the work's crucial observation regarding the difference in the representation matrices for rotations versus reflections is well known. But the fact that they use this for grade discovery seems logical and follows naturally from this observation, and will interest relevant audience.

**Broader Impact Concerns:**

None.

**Claims And Evidence:**

No

**Claims Explanation:**

**The central research question itself is a bit restricted in its scope and the evidence does support it.** However, the claim that leads to the research question does not have enough support/discussion to justify: "the architecture relies on domain experts to map raw coordinates to their corresponding geometric semantics prior to training".

Most grade-based input constructions seem quite intuitive with scalars, positions, velocities mapped usually to a fixed set of grades. So I'm not sure I agree with the claim that only domain experts can set the grade(s) types. More concrete examples and discussions will help support this claim further, given that this is the central basis of the subsequent ideas in this work.

**Requested Changes:**

The research question, while valid, needs the assumption that in this case each channel is assumed to only be of a single grade. In most works (e.g. GCANs, CGENNs, GATr) the inputs are multivectors for many of the experiments and have more than one grade active.


Theoretical results and Proofs can be significantly better organized, currently things are scattered and mixed. Proposition 1 and Theorem 1's proof seems to have been integrated into 1. Please make the proofs separate and formal. Also, I believe Proposition 1 needs an additional assumption, as it would also depend on the distribution of Q. For instance, if Q collapses near zero then essentially the t* being the "only" weight vector condition may be broken. There is also a separate result on the true expected soft weights immediately after which I believe it is not necessary and is answering an irrelevant question. As the authors are looking at "expected soft weight" this is a bit misleading, as that simply then becomes a direct reflection of the probability that the samples "don't" have any reflection pair, which is not a relevant question here. Especially as the authors show for the scalar/pseudo-scalar case immediately after (Appendix F) that the true weights will be 1,0 or 0,1 even if a fraction of the examples have the reflections (ideally please expand F to the general ratio case; and also the vector/bivector case, as it should be straightforward). Because of that result essentially the best weights will be a direct reflection of the type even with a minor fraction of reflection examples as long as they're there. So instead of that result/experiment it would be more useful to comment on the inevitability of the best weights for both the scalar/pseudoscalar and vector/bivector case.


The findings that rotations cannot differentiate between the grade types in the noise-free experiments, and similarly the finding that reflections can, feels largely self-affirming as it is obvious from the model itself that this will hold. So I would suggest that the authors make this sufficiently clear when discussing the results. More rigorous theory on the identifiability in the noise-free setting would be interesting as well.

Apart from this, an honest view on the experiments is that the work operates within a bubble of the main mixed representation result/algorithm and subsequently answers a bunch of questions theory and experiment wise which are precisely within this straightforward linear model setup. What is missing is the types of experiments/datasets where the information presented at the input will actually have these types of differences. For instance, hypothetically a dataset which will have information channels w.r.t velocity and angular momentum will both be in R^3, but one behaves as a vector whereas the other as a bivector. This is important as then some of the experiment setups, such as adding noise to the observations, which can manifest as adding noise to the particle position, or velocities, which affect noise on the bivector side differently than simple Gaussian addition. Or instead, having data which is "pre-augmented" and one needs to guess the transformation from the pair of point-sets etc. where there will be clear matrix observation noise and it will have a certain type of behaviour. This also differentiates between systems where the data comes "pre-probed" versus systems where one can physically probe it by various transformations, and clearly it is expected that the grade discovery results can be different for both (and the computational pipelines as well). Otherwise, it is hard to physically recognize the relevance of the type of observation/matrix noise tested in the experiments. Furthermore, there are other parts that can benefit from a real dataset touch, like the distribution of the inputs and also the distribution of the potential transformations (Also what is the distribution of the orthogonal transformations used in the experiments? I could not find it). So significantly more connections to actual dataset types where such a grade discovery approach holds weight will greatly help the work.

It would also significantly help to organize the experiment sections appropriately, as the experiments are now in one long subsection and as a reader it can be hard to parse the different sub-experiments contained within.

It is a bit surprising to see that the final true type weights don't converge to 1 but to 0.929 rather when a simple additional variable is included in the optimization. My first observation is that why do we want to add more trainable parameters in this case, as the authors just want to identify the grade types of the input channels and once done, use it to design appropriate multivector representations that can be used with downstream Clifford NNs/Transformers. So I'm not sure why this is needed. Also effectively speaking, this feels just a different case of "representation law misspecification" where the scale is not specified. So I don't immediately see the value of this prediction problem in this context, it feels a bit arbitrary. Same for Table 5.


In (6) I'm assuming that the soft weights p_ct is not trained directly via gradient descent but the logits alpha_ct are trainable? Can be made a bit clearer.

---

> ### Author Response · Authors · 2026-07-10
> **Response to Reviewer ErtS**
>
> We thank the reviewer for the review. The comments helped us improve the motivation, scope, theory and experiments. We clarified that ordinary physical grades often do not need discovery, formalized the blockwise assumption, reorganized the proofs and added real geometry and estimated transformation experiments.
>
> Requested Change --> Clarify the motivating claim, since many grade assignments are intuitive.
>
> We agree. The Abstract and Introduction now state that many choices are obvious from physical meaning, e.g. mass/temperature as scalars and velocity/force as polar vectors. The intended use case is now 'ambiguous, derived, learned, or metadata-poor' channels whose coordinate dimension is compatible with more than one Clifford grade. Table 1 separates cases where the diagnostic is usually unnecessary from cases where it is informative, such as torque/angular momentum.
>
> Requested Change --> State the single-grade/blockwise assumption for multivector inputs.
>
> We added Assumption 1 ('Blockwise homogeneous candidates') in Section 3. It states that each diagnostic unit is a coordinate block whose candidate laws act on the same coordinate space. For multivectors, the block partition is assumed known, and the diagnostic can be applied blockwise or to finite block diagonal assignments. We also added Remark 1 and Corollary 2 in Section 4 and the Discussion reiterates that we do not infer the partition or unmix arbitrary dense multivectors.
>
> Requested Change --> Reorganize the theory/proofs, add assumptions on Q and clarify the expected weight result.
>
> We reorganized Section 4 into Theorem 1, Proposition 1, and Corollary 1, with separate proofs in Appendices C, D and E. Proposition 1 now explicitly assumes Pr[det(Q)=-1]>0. Theorem 1 uses an almost sure separation condition in Eq. (5), and Eq. (6) gives the population trace form. We also renamed the former expected weight result to 'Finite-sample probability of observing parity evidence'. The text now states that any positive reflection probability gives a one-hot population optimum.
>
> Appendix D gives the one-hot reflection argument explicitly for vector versus bivector and Appendix E states that the same finite-sample reflection coverage probability applies to both parity-sensitive pairs.
>
> Requested Change --> Make clear that the rotation/reflection result is expected, and strengthen the noiseless identifiability discussion.
>
> We revised the Abstract, Introduction, controlled experiment discussion and Conclusion to state that the SO(3) vs. O(3) parity distinction is classical representation theory. The controlled experiments are now described as sanity checks that the diagnostic reports ambiguity exactly when the laws coincide. After Eq. (6), we state that the theorem is not an optimizer convergence statement, but a null space statement about representation differences under the observed transformations and covariance.
>
> Requested Change --> Add realistic dataset connections, realistic transformation estimation, and specify the transformation distribution.
>
> We added a real geometry MD17 ethanol experiment in Section 5. The channel F_i is treated as a polar vector and tau_i = r_i x F_i as an axial vector/bivector, results are in Table 4 with the full plot in Appendix I. We also added an estimated-Q experiment using Kabsch alignment from noisy paired atom positions. Figure 2 and Table 5 report operational breakdown thresholds as landmark noise and landmark count vary. Appendix F specifies the transformation distribution. The Discussion distinguishes augmentation, physical/simulation probing and registration pairs.
>
> Requested Change --> Improve experiment organization.
>
> We added Table 3 ('Scope and evidence in the experiments') mapping each experiment to the data/model, transformation evidence and supported claim. Section 5 is now split into controlled identifiability, real geometry, estimated transformations and downstream grade-structured O(3)-equivariant message-passing. Supporting plots and implementation checks are in Appendix I.
>
> Requested Change --> Reconsider the minimal trainable module with the extra scalar parameter.
>
> We agree that the learned gain module should not be main evidence. We moved it to Appendix I and label it only as an optimization sanity check. We also removed the original auxiliary equivariant prediction and out of distribution tables from the main argument. The main practical experiment is now the three-layer gradestructured O(3)-equivariant message-passing model in Section 5.
>
> Requested Change --> Clarify whether p_{ct} or logits α_{ct} are trained.
>
> We clarified this in Section 3 around Eq.(4): α_{ct} are unconstrained trainable logits, and p_{ct} is their softmax normalized value, not an independent unconstrained parameter.
>
> We hope these changes address the concerns and make the diagnostic’s contribution and assumptions clearer.